# The Ambiguous Correlation of *Blautia* with Obesity: A Systematic Review

**DOI:** 10.3390/microorganisms12091768

**Published:** 2024-08-26

**Authors:** Warren Chanda, He Jiang, Shuang-Jiang Liu

**Affiliations:** 1State Key Laboratory of Microbial Technology, Shandong University, Qingdao 266237, China; 2Pathology and Microbiology Department, School of Medicine and Health Sciences, Mulungushi University, Livingstone P.O. Box 60009, Zambia; 3State Key Laboratory of Microbial Resources, and Environmental Microbiology Research Center (EMRC), Institute of Microbiology, Chinese Academy of Sciences, Beijing 100101, China; 4University of Chinese Academy of Sciences, Beijing 100049, China

**Keywords:** *Blautia*, probiotics, obesity, gut microbiota, *Blautia*–obesity correlation

## Abstract

Obesity is a complex and multifactorial disease with global epidemic proportions, posing significant health and economic challenges. Whilst diet and lifestyle are well-established contributors to the pathogenesis, the gut microbiota’s role in obesity development is increasingly recognized. *Blautia*, as one of the major intestinal bacteria of the Firmicutes phylum, is reported with both potential probiotic properties and causal factors for obesity in different studies, making its role controversial. To summarize the current understanding of the *Blautia*–obesity correlation and to evaluate the evidence from animal and clinical studies, we used “*Blautia*” AND “obesity” as keywords searching through PubMed and SpringerLink databases for research articles. After removing duplicates and inadequate articles using the exclusion criteria, we observed different results between studies supporting and opposing the beneficial role of *Blautia* in obesity at the genus level. Additionally, several studies showed probiotic effectiveness at the species level for *Blautia coccoides*, *B. wexlerae*, *B. hansenii*, *B. producta*, and *B. luti*. Therefore, the current evidence does not demonstrate *Blautia*’s direct involvement as a pathogenic microbe in obesity development or progression, which informs future research and therapeutic strategies targeting the gut *Blautia* in obesity management.

## 1. Introduction

Obesity is a complex and multifactorial disease characterized with a body mass index (BMI) of over 30 kg/m^2^. It is influenced by a combination of factors, including intestinal microbiota, genetic background, diet and lifestyle, and environment. Being a serious threat to public health, obesity increases the risk of various health conditions (Figure 1A), such as obstructive sleep apnea, which is accompanied by hypertension, insulin resistance, systemic inflammation, visceral fat deposition, and dyslipidemia [1,2]; stroke; hypercholesteremia; diabetes mellitus [3,4]; non-alcoholic fatty liver disease [5]; osteoarthritis [6]; asthma and chronic pulmonary disease [7]; infertility in both men and women [8,9]; depression [10]; and various types of cancers, such as breast, colon, liver, ovarian cancers, etc. [11]. As reported in 2021, over 40% of adults globally are overweight or obese, with projections indicating that, by 2030, 20% of the adult population will be affected [12,13,14]. In China, the rates of overweight and obesity among adults (≥18 years old) have reached 34.3 and 16.4%, respectively [15]. 

The commonly practiced solutions to control obesity and body weight are bariatric surgery and lifestyle management. While lifestyle-based weight loss strategies that combine diet, exercise, and behavioral changes (e.g., reducing sedentary behavior) can initially be successful, they often lead to weight regain due to metabolic adaptations [16]. Thus, there is a growing interest in pharmacological solutions, particularly those targeting the gut microbiota—the diverse community of microorganisms residing in the gastrointestinal tract. These microorganisms, which play crucial roles in nutrient metabolism and immune function, influencing the overall health and disease [17,18,19]. Research has shown that obesity is associated with reduced gut microbiota diversity and has identified key taxa, including *Blautia*, that exhibit significant alterations in obese individuals [20,21]. These alterations can be affected by underlying conditions such as diabetes and metabolic syndrome [21,22]. Additionally, the gut microbiota impacts energy metabolism and the production of short-chain fatty acids (SCFAs)—fatty acids with fewer than six carbon atoms, primarily produced by the fermentation of dietary fibers by gut bacteria. SCFAs, such as acetate, propionate, and butyrate, which help regulate energy balance and inflammation, contribute significantly to host metabolism and health [20].

*Blautia*, a genus within the *Lachnospiraceae* family, comprises 3–11% of the human intestinal microbiota [23,24,25]. Although only a few child taxa (23 out of 53 classified species) are validly published (Appendix A), *Blautia* is commonly detected in human fecal samples from both healthy and diseased individuals (Figure 1B). Some studies suggest that *Blautia* may positively influence the lipid metabolism and reduce visceral fat accumulation [25,26,27]. However, its role in obesity remains ambiguous, with conflicting evidence regarding its impact on health [27,28,29,30,31,32,33]. This highlights the need for a comprehensive evaluation of *Blautia*’s complex interaction with the host to understand its specific role in metabolic disorders such as obesity.

This systematic review aims to summarize the recent findings on *Blautia*’s role in obesity. We analyzed 170 research articles to explore the abundance of *Blautia* in the gut microbiome of obese individuals concerning treatment and lifestyle interventions. Additionally, we explored how *Blautia* populations respond to any treatments and lifestyle changes in obese individuals, examining associations between changes in *Blautia* abundance and the efficacy of employed interventions in managing obesity. Lastly, we discuss the potential of *Blautia* as a probiotic or source of beneficial metabolites for promoting health.

**Figure 1 microorganisms-12-01768-f001:**
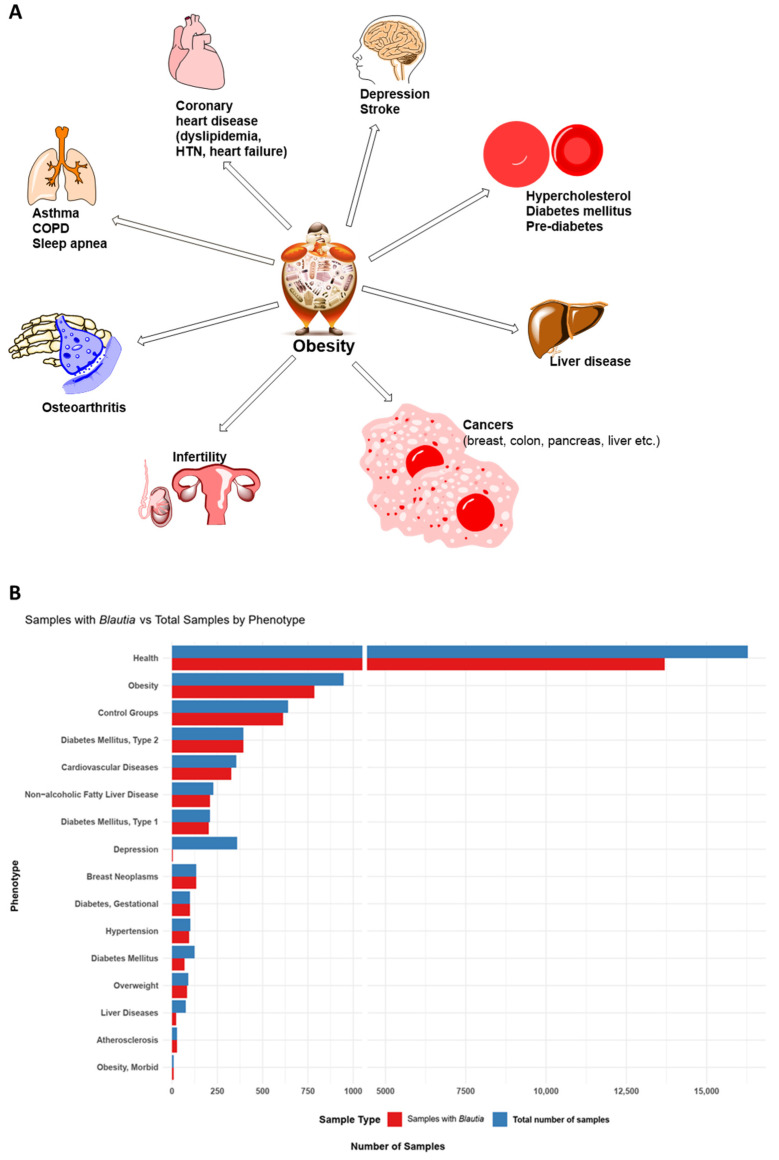
The associations of obesity with some concomitant diseases (**A**), and the prevalence of *Blautia* in phenotypes (**B**). Panel B was generated from data compiled by the GMRepo (Gut Microbiota Repository) database. The prevalence of *Blautia* (red) in the total number of samples processed per phenotype (blue) is shown [34,35] (https://gmrepo.humangut.info/taxon/572511, accessed on 15 July 2024).

## 2. Methods

### 2.1. Literature Search Strategy

We utilized PubMed and SpringerLink search engines as our primary tools, employing a key Boolean phrase “*Blautia*” AND “Obesity” to identify relevant articles for our systematic review following the Preferred Reporting Items for Systematic Reviews and Meta-Analyses (PRISMA) guidelines [36]. We considered animal models and clinical research papers published from January 2013 to March 2024, while review articles were excluded. 

### 2.2. Study Selection

Two independent reviewers guided by the inclusion and exclusion criteria conducted an initial assessment of the titles and abstracts of all the retrieved articles. Any conflicts that emerged over research study selections were resolved by discussion with the authors. The PICOS (Population, Intervention, Comparison, Outcome, Study design) framework was used to define the eligibility criteria.

PICOS criteria:Population: Animal models and human subjectsIntervention: Any treatment or lifestyle interventions affecting gut microbiotaComparison: Factors inducing gut microbiota changes, including Firmicutes/Bacteroidetes (FB) ratio, molecular methods used, treatment period, and subjects’ countryOutcome: Changes in Blautia abundance and obesity statusStudy design: Original research articles

### 2.3. Data Extraction

With the aim to summarize the understanding of the associations and mechanism between *Blautia* and obesity, we particularly considered research articles that reported variables covering the effect of medical treatment or lifestyle on the status and dynamics of *Blautia*, factors that induced gut microbiota changes (including FB ratio, molecular method used, treatment period, and country of subjects), and variables related to the associations and mechanisms between *Blautia* and obesity (or overweight) were independently extracted by the authors using a standardized data extraction form in Microsoft Excel 2021.

### 2.4. Bias Evaluation

All studies meeting the inclusion criteria were considered for data extraction, irrespective of their quality. The potential risk of bias was assessed by evaluating three domains (selection bias, performance bias, and detection bias), following O’Connor and Sargeanti’s extraction [37] from the Cochrane Handbook of Systematic Reviews of Interventions and adapted for animal studies in preclinical medicine [38]. Two independent reviewers performed the bias evaluation, and any disagreements were resolved through discussion or by consulting a third reviewer. 

### 2.5. Data Synthesis

Descriptive statistics and association tests, including Spearman’s correlation and/or chi-square tests, were performed using IBM SPSS Statistics version 20 software. 

## 3. Results

### 3.1. Studies Overview

Following a comprehensive search of the SpringerLink and PubMed databases, 1378 research articles published between January of 2013 and March of 2024 were found and retrieved. Of these, 170 articles that satisfied the inclusion criteria were chosen for in-depth analysis after a thorough screening process (Figure 2A). Of the above 170 articles analyzed, varying proportions of the selected variables of interest were observed (Figure 2B). Among these variables, the publication year highlighted an increasing pattern of *Blautia* and obesity-related studies until 2022, then a drop in 2023, with mice, children, and adults being the commonest study subjects. Gut microbiota alterations in relation to obesity were mostly induced with prebiotic supplementation. To summarize the roles that *Blautia* played in obesity-related gut microbiota studies, it could be descripted as either “good” or “bad”, depending on whether the *Blautia* abundance was negatively or positively associated with obesity and other side symptoms, respectively. Briefly, *Blautia* was described as “good” in >60% of the studies evaluated (Figure 2B). This was mostly attributed to the increased level of *Blautia* detection in interventional studies, *Blautia*’s ability to produce metabolites (such as SCFAs) that alleviated obesity and related indices, and the observed suppression of obesity in some subjects compared to the control group. Additionally, the use of *Blautia* as probiotic agents had significant ameliorating effects, suggesting its potential role in promoting health (Appendix A). Interestingly, the Firmicutes-to-Bacteroidetes (FB) ratio was lowered in the majority of obese cases (Figure 2B), highlighting the importance of understanding microbial interactions and their implications for host health. We observed that the FB ratio may decrease in circumstances where *Blautia* increases, especially with prebiotic supplementation [39,40,41], diet alteration [42], or surgery [43,44,45]. Additionally, an increasing FB ratio was associated with either an increase or decrease in obese subjects regardless of whether the *Blautia* status was “good” or “bad” (Figure 2B and Appendix A), indicating that the FB ratio varies and cannot be used as a reliable predictor of obesity, since alterations at the phylum level do not always translate to all members at the family or genus level. 16S rRNA sequencing was a predominant molecular method used with a focus on the V3–V4 regions. However, none of the studies sequenced the full length (~1500 bp) of the 16S rRNA gene, except for the studies by Sun et al. [46] and Gómez-Pérez et al. [42], which sequenced the longest regions (V3–V9 and V2–V9, respectively) (Figure 2B and Appendix A). Finally, a suitable treatment period is crucial when examining the impact of treatment regimens with fewer adverse effects [47,48]. Most studies assessed their therapies for 12 weeks, followed by 8 and 4 weeks (Figure 2B). Additionally, the majority of the pooled studies were conducted in China, followed by Spain, Japan, and the United States. Other geographical regions had fewer studies (<2%), but when combined, they produced a well-represented geographical distribution (Figure 2B).

Since most studies induced changes in the gut microbiota through medical treatments (such as bariatric surgery and fecal microbiota transplantation) and lifestyle management (including diet, exercise, prebiotics, and probiotics), and to explore the dynamics of *Blautia* populations in obese individuals concerning these interventions, the results will be organized according to these criteria.

### 3.2. Status and Dynamics of Blautia Population during Medical Treatment and Lifestyle Managements in Obese Individuals

#### 3.2.1. Surgery

Bariatric surgery is often regarded as the primary therapy for obese and overweight individuals. The most common procedures include sleeve gastrectomy (SG) and Roux-en-Y gastric bypass (RYGB) [49]. RYGB is excellent at reducing weight, but bypassing the large section of the stomach and the upper part of the small intestine reduces the absorption of essential vitamins and trace elements, causing nutritional deficiencies and, consequently, impacting the gut microbiota [50]. On the other hand, SG is a restrictive procedure that reduces the stomach size to about 15–25%, which has less impact on the nutritional deficiency and gut microbiota [50]. The data show that SG but not RYGB may be associated with an improvement in obesity due to an increase in the abundance of *Blautia* post-surgery, which significantly reduces host weight gain, inflammatory factors, and visceral fat accumulation (Appendix A) [43,44,45,49,51,52]. Additionally, an increase in ursodeoxycholate, which may improve liver health and prevent gallstone disease, was observed [51,53]. Therefore, the increased abundance of *Blautia* in the gut microbiota after bariatric surgery may also serve as a potential indicator of successful weight loss and improved metabolic health [45]. 

#### 3.2.2. Diet 

The ketogenic diet (approximately 10% carbohydrates, 70% fats, and 20% protein) and the Mediterranean diet are thought to be beneficial for weight management in terms of changes in the gut microbiota (as discussed in Ref. [16]). Evaluating the effect of these diets on *Blautia* may help predict its involvement in health and obesity. Among the clinical studies analyzed (Table 1), a decreased calorie intake enhanced the amount of *Blautia*, which had a positive influence on health. A combination of diet and exercise [54], medium-protein and low-fat diets [55], the Mediterranean diet and exercise [42], improved ketogenic diet and exercise [56], and hypocaloric hyperproteic diet [57] enhanced the *Blautia* levels, leading to the amelioration of obesity and its related indices. However, a study that paired an enhanced ketogenic diet with exercise discovered that *B. obeum* had a favorable correlation with VFA, whereas *B. producta*, *B. hansenii*, *B. wexlerae*, and *Blautia* sp. CAG257 showed negative associations (Table 1, [56]). This observation may be due to a species- and/or strain-specific response of *Blautia* that may not be generalized to the entire genus. Together, these studies underscore the importance of diet composition for boosting the potential probiotic usage of strain-dependent *Blautia* levels to alleviate obesity.

#### 3.2.3. Exercise

Regular exercise is known to lower the incidence of metabolic diseases [61]. Despite exercise being recommended for improving health in overweight or obese individuals, its specific effects on the gut microbiota composition, particularly with *Blautia*, remain unclear. Quiroga et al. [62] emphasized that exercise enhances the abundance of *Blautia*, which correlated with reduced adiposity, improved lipid profiles, and decreased inflammation in obese children. Additionally, short-term and high-intensity interval training has been shown to increase the abundance of bacterial species such as *Coprococcus*_3, *Blautia*, and *Dorea*, correlating with markers of insulin sensitivity in overweight individuals but not affecting the gut bacterial diversity in lean or overweight men [61]. Whether the observed increase in *Blautia* abundance and exercise exerted a synergistic effect on the disease outcomes requires further investigation. 

Furthermore, aerobic exercise has been found to increase circulating levels of endocannabinoid agonists, including anandamide and 2-arachidonoylglycerol, which play crucial roles in appetite regulation, metabolism, stress modulation, and inflammation [63,64], thereby reducing the occurrence of obesity [65]. Palmitoylethanolamide (PEA), similar to cannabidiol in its pharmacodynamic properties, inhibits the degradation of these endocannabinoid agonists [66,67] and has been reported to associate with increased *Blautia* abundance and alleviation of anhedonia/amotivation [68]. Given that both *Blautia* and cannabidiol/PEA help preserve the gut barrier integrity [69,70] and reduce colonic permeability [67], it is plausible that *Blautia* is involved in regulating the endocannabinoid levels in conjunction with exercise.

#### 3.2.4. Probiotics

Probiotics are defined as living microorganisms that, when administered in adequate amounts, confer health benefits to the host, often by modulating the gut microbiota. There is a paucity of studies on the interaction of *Blautia* with other potential probiotic strains in obesity management. Zhao et al. [71] found a reduced level of *Lactobacillus*, *Allobaculum*, and *Blautia* in the high cholesterol diet group compared to the normal diet group in C57BL/6 male mice. However, treatment with the lactic acid bacterium, *Lactiplantibacillus plantarum* WLPL21, restored these taxa. The study also discovered a positive correlation of serum and liver cholesterol with *Desulfovibrio* and *Lachnoclostridium*, while a negative correlation was noted with *Blautia*, *Allobaculum*, and *Lactobacillus*. Similarly, Jing et al. [72] found that the administration of *Bacillus* sp. DU-106 in C57BL/6J mice restored the abundance of *Blautia, Ileibacterium, Faecalibaculum, Faecalibacterium*, and unidentified_*Lachnospiraceae*, which were reduced by a high-fat diet. However, a 12-week intervention with *Lactobacillus salivarius* Ls-33 among obese adolescents detected no significant alteration of the *Blautia coccoides*–*Eubacterium rectale* group in patients before and after probiotic intake [73]. Although the detection method (RT-qPCR) was less sensitive due to its multispecies amplification, the results indicate that *Blautia* may have a specific role in disease management (Ref. [73], Appendix A). In support of this, the administration of *B. hansenii* improved obesity in C57BL/6J mice by increasing its abundance [74]. These findings imply that, while *Blautia* plays a role in cholesterol metabolism, its effects may be enhanced by interspecies interactions with the gut microbial community.

#### 3.2.5. Prebiotics

Prebiotics are a category of non-digestible food components that selectively stimulate the growth and/or activity of beneficial gut bacteria. Numerous in vivo and clinical studies have shown that prebiotics can promote host health by enhancing the growth of beneficial gut bacteria, which subsequently regulates the balance of the gut microbial flora, heightens the gut resistance to diseases, and influences the host immunity [75,76]. The use of prebiotics such as arabinoxylan, berberine, celery juice, d-arabitol, inulin, resveratrol polyphenol, silybin, yacon, etc. (Figure 3A, green box) elevated the levels of *Blautia*, as well as some other SCFA-producing bacteria such as *Bacteroides* and *Akkermansia*, while inhibited lipopolysaccharide (LPS)-producing and proinflammatory bacteria, resulting in a significant increase of SCFAs, especially butyric acid, reduced serum LPS levels and inflammation, and regulating visceral fat and hepatic lipid metabolism (Appendix A) [41,77,78]. Conversely, the administration of prebiotics such as adzuki beans, mung bean seed coat, moringa oleifera polysaccharides, quinoa, vitamin D, etc. (Figure 3A, red box) improved weight management but decreased the abundance of *Blautia* (Appendix A). These findings suggest *Blautia*’s potential probiotic role in obesity, with prebiotics having varied impacts on its abundance, highlighting the importance of prebiotic selection when targeting *Blautia* for probiotic effects.

#### 3.2.6. Fecal Microbial Transplant (FMT)

Fecal microbiota transplantation (FMT) is a medical procedure that involves transferring fecal microorganisms from a healthy donor into the gut of a recipient to treat specific diseases [80,81]. FMT is associated with mixed results in obesity management, with some promising effects, especially in improving insulin sensitivity, but with limited impact on weight loss [80,82]. Participants with BMI ≥ 35 kg/m^2^ administered autologous FMT or from a healthy lean donor had a notable increase in the abundance of *Bacteroides* and *Blautia*, coupled with reduced serum levels of branched-chain amino acids (isoleucine and leucine), decenoylcarnitine, and fecal phenylacetic acid [81]. Elevated serum levels of branched-chain amino acids and decenoylcarnitine are recognized markers of obesity and insulin resistance, attributed to the activation of the mammalian target of rapamycin complex 1 (mTORC1) [83,84,85]. Conversely, changes in the fecal phenylacetic acid levels may signify alterations in the gut microbial metabolism associated with obesity. A decrease in its concentration suggests a favorable shift towards a healthier gut microbiome composition and function [81,86]. While FMT had no effect on weight loss before bariatric surgery [80], the increased abundance of *Blautia* and decreased metabolites associated with obesity and insulin resistance in serum underscore the importance of identifying and using selected species like *Blautia* with probiotic potential. 

### 3.3. Association between Blautia and Obesity

*Blautia* species have the potential to impact obesity through multiple pathways, such as modifying energy metabolism, reducing inflammation, and influencing the composition of the gut bacterial community [28,87,88]. Improvements in glucose metabolism, such as increased insulin sensitivity, which controls the blood glucose levels, are among the suggested mechanisms [29,74]. Furthermore, *Blautia* can affect the metabolism of fatty acids, lowering the accumulation of harmful lipid species in the liver and altering the composition of circulating bile acids, which are involved in fat absorption [28,29,41]. Moreover, *Blautia* species have the ability to regulate immunological responses and support the integrity of the intestinal barrier, which may positively impact metabolic health and reduce systemic inflammation [28,29,41].

#### 3.3.1. Association of Blautia and Obesity in Children and Adolescents

Childhood obesity is a prognostic marker of adult obesity, and the prevention of obesity in children may help to address the epidemic [10]. The diversity of gut microbiota in children can be influenced by maternal health status (obese/overweight or lean), mode of delivery (vaginal vs. cesarean section), and feeding practices (exclusive breastfeeding or alternative methods, with formula-fed infants showing an increased risk of overweight compared to breastfed infants [89]). Consequently, offspring born to obese mothers are presumed to have an elevated risk of obesity, potentially linked to the transmission of obesogenic microbiota. However, despite children born to obese mothers exhibiting a lower carbohydrate content, reduced levels of *Blautia* and *Eubacterium*, and increased levels of *Parabacteroides* and *Oscillibacter*, no significant associations were found between the gut microbial community structure and environmental factors such as delivery mode, breastfeeding duration, or maternal and infant antibiotic exposure [90]. The reduced levels of *Blautia* may be impacted by other factors besides dietary composition, as there is a lack of association between protein, fat, carbohydrate, or fiber intake and *Blautia* abundance [25].

While *Blautia* species are associated with obesity in children, as noted by the “good” and “bad” status in studies involving children (Appendix A), the relationship is complex and may depend on factors such as age, inflammation, and diet. One study indicated the presence of *Blautia* in a meconium sample that disappeared within the first 5 months of the child’s life and reappeared in month 6 (Figure 3B, data extracted from the Supplementary Dataset of Ref. [79]). Some *Blautia* species (e.g., *B. luti* and *B. wexlerae*) are reported to have anti-inflammatory and anti-obesity effects, while others (e.g., *B. producta*) are negatively correlated with inflammatory markers like LPS binding protein (LBP), showing an opposite correlation with obesity [88]. 

*Blautia*’s role in ameliorating obesity would also be observable in adolescents. However, one study found varying levels of *B. faecis* and *B. wexlerae* in normal weight healthy donors compared to obese individuals, with *B. producta* associated with severe overall obesity in Italian adolescent subjects [91]. The significant level of *Blautia* abundance was observed in adolescent subjects who consumed *Lactobacillus salivarius* Ls-33 for 12 weeks compared to the placebo group [73]. This further complicates the association of *Blautia* with obesity, indicating the need for further investigation. Obesity, often resulting from a long-term imbalance between energy intake and expenditure, has been linked to energy-harvesting microbes like *Blautia* and *Bacteroides* [16]. While calorie restriction is a common practice in weight management, one study showed an increased relative abundance of *Blautia* in obese adolescents. However, this increase did not significantly impact the metabolic processes of the gut microbiome, such as energy harvesting or nutrient absorption, which might have been expected given *Blautia*’s role in carbohydrate metabolism [92]. 

These findings collectively suggest that *Blautia* depletion is associated with obesity and insulin resistance, and modulation after physical activity interventions restores it to resemble that of healthy children [93]. The beneficial effects of *Blautia* in children and adolescents may include, but are not limited to, contributions to carbohydrate metabolism and SCFA production, as well as anti-inflammatory, antioxidative, and anti-infective properties [26].

#### 3.3.2. *Blautia* and Obesity among Adult Subjects

In studies involving adult human subjects, *Blautia* had been categorized as “good” in 66% (*n* = 35/53) of the reports (Appendix A), regardless of the FB ratio, genome sequencing, and treatment methods. Based on subjects’ geographical locations, all studies found *Blautia* had beneficial effects (>50%), except those from Australia, Poland, and Sweden with one study each and Finland with three studies (Figure 4 and Appendix A). An increasing pattern of studies describing *Blautia* as “good” was seen from the years 2018–2023, with all gut microbiota manipulation (inducer type) showing a >50% description of *Blautia* as “good”. Furthermore, a negative association of *Blautia* with obesity was seen in 97.2% (*n* = 35/36) of studies, while 100% (*n* = 17/17) reported a positive association (Figure 4 and Appendix A).

Significant differences in the gut microbial diversity between obese and non-obese individuals suggest a potential link between bacterial composition and obesity-related diseases. Kasai et al. [94] identified distinct bacterial species in Japanese obese subjects, including *B. hydrogenotorophica*, *Coprococcus catus*, *Eubacterium ventriosum*, *Ruminococcus bromii*, and *R. obeum*, whereas non-obese subjects showed *Bacteroides faecichinchillae*, *Bac. thetaiotaomicron*, *B. wexlerae*, *Clostridium bolteae*, and *Flavonifractor plautii*. This disparity underscores the varied associations of *Blautia* species with obesity in adults. Therefore, it is critical to identify specific species of *Blautia* with beneficial effects. For instance, *B. hansenii* and *B. producta* were negatively associated with VFA in a longitudinal study involving 767 Japanese subjects [27]. In another study, *B. wexlerae* demonstrated anti-obesity and anti-diabetic properties in Japanese adults, which was evidenced by its efficacy in reducing obesity and diabetes symptoms in mouse models [28]. The reported efficacy was associated with a unique metabolic pathway involving amino acid derivatives like S-adenosylmethionine and l-ornithine, as well as carbohydrate metabolism-producing SCFAs [28]. SCFAs activate G-protein-coupled receptors (GPCRs) in human adipocytes and colon cells enhancing insulin sensitivity through GPR 43 expression in adipose tissue [74,95,96,97]. Moreover, an increased abundance of *Bacteroides* and *Blautia* was associated with reduced serum levels of isoleucine, leucine, decenoylcarnitine, and fecal phenylacetic acid, which are recognized markers of obesity and insulin resistance [81,83,84,85].

## 4. Discussion

This comprehensive review aims to address the ongoing debate surrounding the role of *Blautia* in obesity, a topic that often elicits divergent opinions among scholars. The question of whether *Blautia* should be classified as a “friend” (beneficial) or “foe” (detrimental) in health and disease remains contentious, with many authors presenting biased perspectives shaped by their own research findings. In an effort to provide a more balanced assessment, we conducted a thorough examination of both in vivo and clinical data in accordance with PRISMA guidelines, specifically focusing on *Blautia*’s distinct association with obesity. Our objective was to determine whether *Blautia* should be viewed as a facilitator or a mitigator of disease risk, moving away from a simplistic “guilt by association” approach to a more nuanced evaluation of its impact.

Despite the scarcity of studies specifically evaluating *Blautia*’s role in obesity, examining its population dynamics in relation to gut microbiota manipulations can provide insights into whether *Blautia* should be considered a “friend” or “foe” in obesity management. Our comprehensive analysis of data from the GMRepo database [34,35] revealed significant variability in *Blautia* prevalence among the samples (see Figure 1B), indicating a potential association with host phenotypes. However, the presence of *Blautia* alone does not indicate causation, underscoring the need for experimental validation. The increased *Blautia* in obese adolescents without significant gut microbiota metabolic changes further highlights this complexity [92]. Certain bacteria can influence host conditions by secreting virulent or metabolic factors. Conversely, the absence of these factors may indicate a lack of participation in the observed phenotypes. Therefore, it is crucial not to draw simplistic conclusions, as the mere presence of bacteria does not necessarily imply a direct role in the condition. While the reviewed studies cannot establish a direct causal link between *Blautia* and the observed outcomes, they suggest a potential involvement in the development and prevention of obesity (Figure 5). Further investigations are needed to explore the causal relationship between *Blautia* and its effects, providing a deeper understanding of its role in these physiological mechanisms.

*Blautia* is a noteworthy genus of bacteria that has an impact on children’s growth and health that can last into adulthood. Extended periods of exclusive breastfeeding are linked to early life colonization of *Blautia* species, including *B. luti* and *B. wexlerae*, which may lay the foundation for a healthy gut microbiota [99]. Because its depletion is associated with childhood obesity and insulin resistance [29], it is plausible to link *Blautia* with metabolic health. Moreover, *Blautia* species have anti-inflammatory properties and are involved in the metabolism of carbohydrates, leading to the production of beneficial metabolites such as SCFAs [29,69,99]. The abundance of *Blautia* increases as children grow into adults [100], suggesting a fundamental role in mature gut function. This change may guard against obesity and prediabetic conditions while also assisting in the maintenance of gut immunological homeostasis [29,69]. Thus, with long-term effects that extend into adulthood, *Blautia* is important for children’s health and may be targets for microbiota-based therapies aimed at managing or preventing obesity and associated metabolic disorders. Their abundance and presence in the gut microbiome may also function as markers of metabolic health.

The relationship between obesity and metabolic syndrome has been extensively studied in regard to the Firmicutes–Bacteroidetes (FB) ratio; higher ratios have been related with obesity, while lower ratios have been linked to type 2 diabetes [101,102]. However, the FB ratio was not consistent with the *Blautia* species, which corroborated with previous findings [103]. Moreover, studies have shown that increasing the abundance of certain probiotic bacteria, including *Blautia* species, through probiotic and prebiotic supplementation may contribute to obesity management [28,74,104,105]. In addition to probiotic and prebiotic interventions, dietary modifications such as increased fiber intake have been shown to increase *Blautia* abundance in human participants [25,92].

*Blautia* species may exert their anti-obesity effects through various mechanisms, including altering energy metabolism [92], exerting anti-inflammatory effects [26,88], and changing the composition of the gut bacterial environment [87]. This means that methods to enhance *Blautia* abundance may be a viable strategy for assisting weight management initiatives, even though additional research is required to completely understand the potential of *Blautia*-targeted therapies.

Several limitations of this study must be acknowledged. Firstly, the scarcity of studies specifically focusing on *Blautia*’s role in obesity required us to include studies that reported alterations in *Blautia* among other gut microbiota taxa. Attributing specific functions to *Blautia* was challenging, because many studies also involved other bacterial species. Another limitation is that our review protocol was not registered with databases like PROSPERO, although we adhered to the PRISMA guidelines (Figure 2A). Additionally, our review only included articles from two databases, PubMed and SpringerLink, and was limited to English-language studies, which may have excluded relevant research. To mitigate study selection bias, at least two independent reviewers participated in the study screening and data extraction process. Despite these limitations, our findings contribute to a broader understanding of the link between *Blautia* and obesity and lay the foundation for future research. It is important to consider the implications of these results for practice, policy, and further investigation.

In conclusion, applying the concept of “guilt by association” to *Blautia*’s role in obesity may be premature, as the current evidence does not demonstrate *Blautia*’s direct involvement as a pathogenic microbe in the development or progression of obesity. Our study indicates that *Blautia* populations may change with different gut microbiota manipulations in obese individuals, potentially improving or exacerbating obesity. This variability may be due to species- or strain-specific effects, which should be considered when evaluating *Blautia’s* role in obesity. Some *Blautia* strains are capable of producing beneficial metabolites and inhibiting harmful products to the host (Figure 5). Notably, certain species within the *Blautia* genus, such as *B. coccoides*, *B. wexlerae*, *B. hansenii*, *B. producta*, and *B. luti*, have shown probiotic effectiveness (Figure 5). This calls for advanced sequencing methods, like shotgun sequencing or complete 16S rRNA gene sequencing, to provide better insights into *Blautia* species in diseased phenotypes and may help assigning an appropriate status to *Blautia* in obesity and related metabolic disorders. Additionally, more in vitro experiments are needed to explore *Blautia*’s functions for a better understanding of their roles in obesity. Lastly, *Blautia*’s presence can be influenced by interactions with other gut microbiota members, dietary components, or prebiotics, emphasizing the importance of careful dietary selection to appreciate its function in obesity.

## Figures and Tables

**Figure 2 microorganisms-12-01768-f002:**
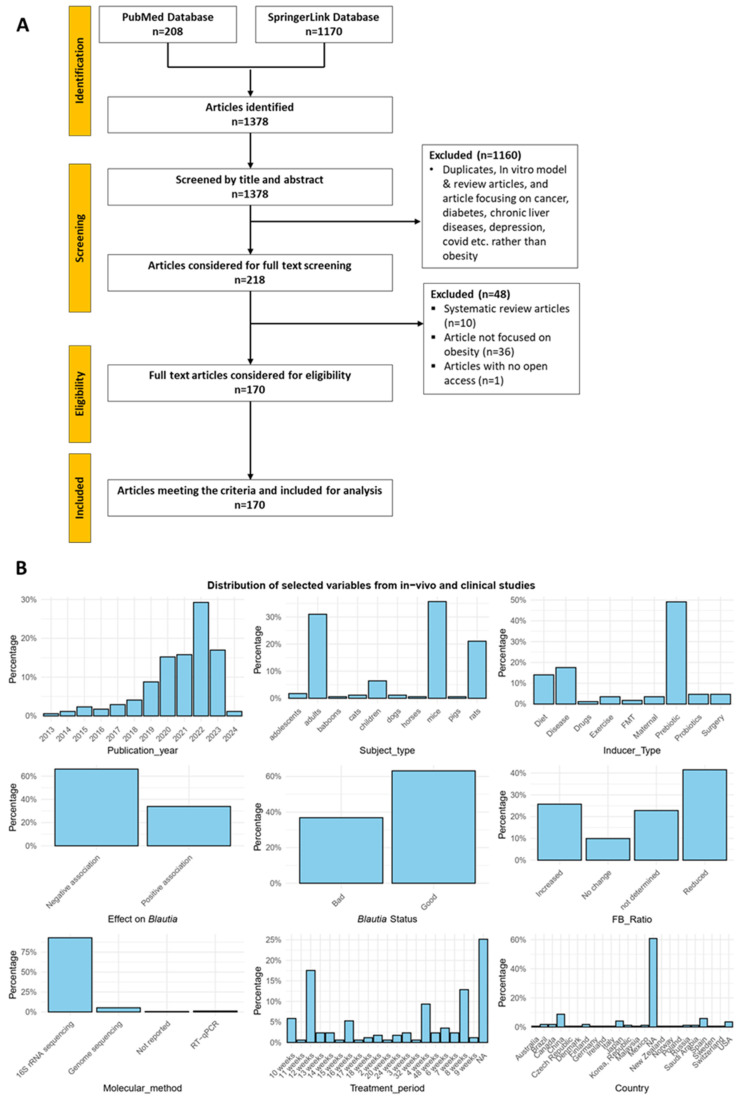
Study selection flowchart following the PRISMA guidelines (**A**), and proportions of the selected variables from in vivo and clinical studies ((**B**), *n* = 170).

**Figure 3 microorganisms-12-01768-f003:**
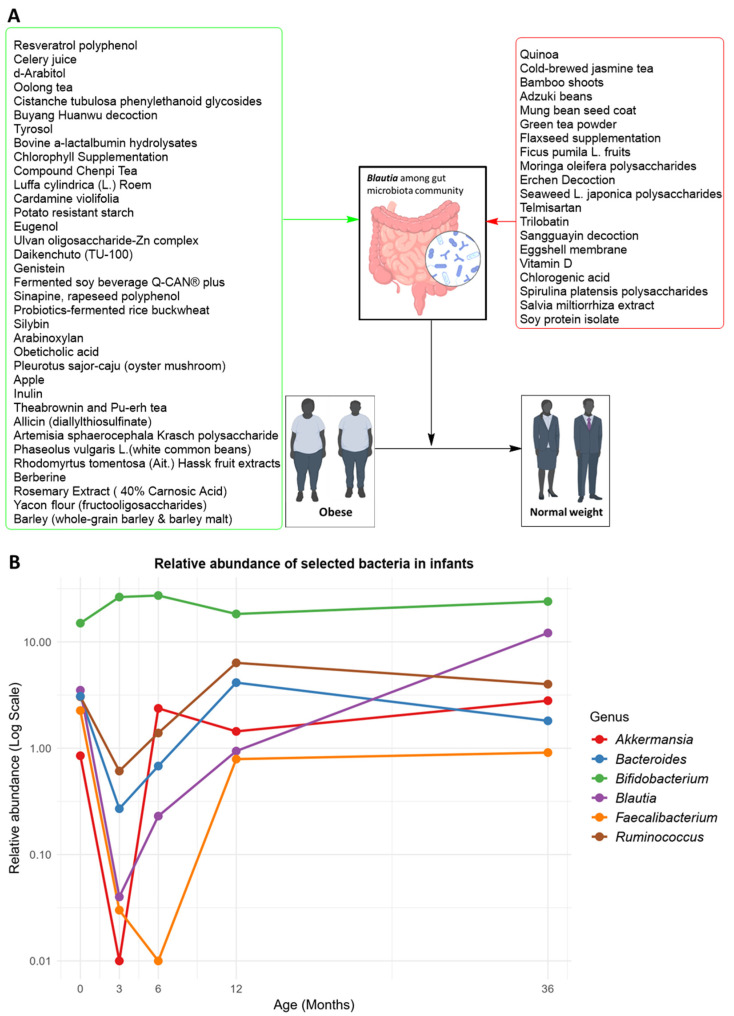
(**A**) Effect of prebiotics on *Blautia*, and (**B**) the relative abundance of selected bacteria from the most and least dominant bacterial taxa in infant fecal samples at ages 0 (first pass meconium), 3, 6, 12, and 36 months. Panel A shows the types of prebiotics in green and red boxes that increased (green arrow) and reduced (red arrow) the abundance of *Blautia* in their respective animal and human subjects that subsequently improved obesity. Note that the obesity-ameliorating effects of some prebiotic products in both green and red boxes were observed in in vivo studies, but the human pictures in the figure were used as representations of obesity and normal weight. Panel B shows a line graph indicating the significant presence of *Blautia* in meconium samples, which decreased substantially at month 3, followed by a steady increase from month 6 to month 36 (This figure was generated from the Supplementary Dataset of Ref. [79].).

**Figure 4 microorganisms-12-01768-f004:**
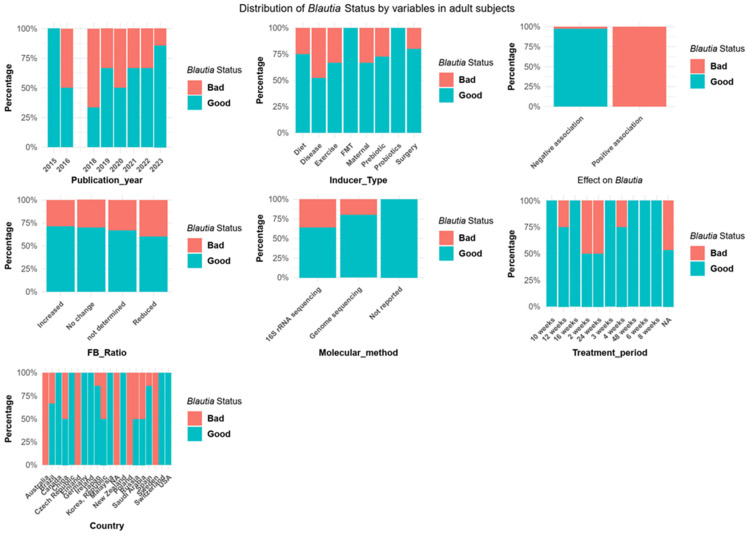
*Blautia* status in adult participants (*n* = 53).

**Figure 5 microorganisms-12-01768-f005:**
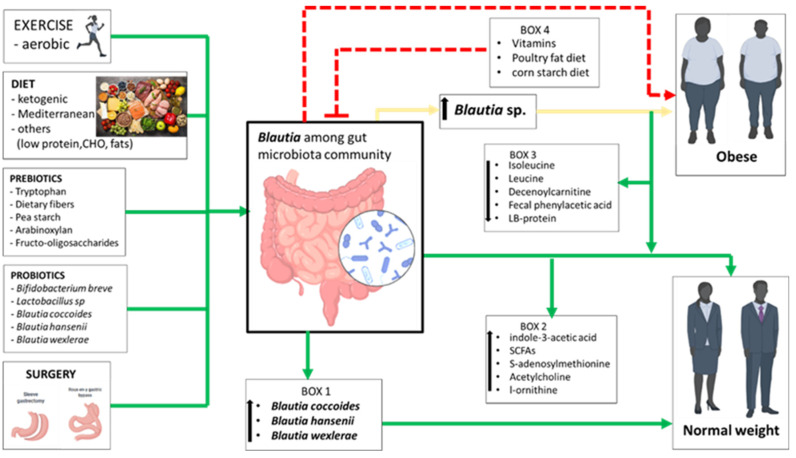
Factors influencing the abundance of *Blautia* in the gut. The proliferation of members of the genus *Blautia* or specific species (Box 1) and the reduction of certain compounds (Box 3), either independently or synergistically, produce important metabolites (Box 2) that may help alleviate obesity (green lines). However, unclear events lead to an increased abundance of some un-speciated *Blautia* associated with obesity (yellow lines). Additionally, factors inhibiting *Blautia* abundance (Box 4) may exacerbate obesity (red dotted lines). Note that the effect of a cornstarch diet on *Blautia* has only been reported in cats [98], so caution should be taken when extrapolating these findings to humans. Also, note that the obesity-ameliorating effects of some interventions were observed in in vivo studies; the human images in the figure are used as representations of obesity and normal weight.

**Table 1 microorganisms-12-01768-t001:** *Blautia* response in gut microbiota changes due to diet in adult subjects.

Refs.	Inducer	Effect on *Blautia*	Status	FB Ratio	Method Used	Treatment Period	Country	*Blautia* Related Findings
Jie et al. 2021 [54]	dietary and exercise	(−)	Good	nd	metagenomic shotgun sequencing	24 weeks	China	*B. wexlerae* was the strongest predictors for weight loss when present in high abundance at baseline.
Cuevas-Sierra et al. [55]	Medium protein & low-fat diets	(−)	Good	nd	16S rRNA, V3–V4 region sequencing	16 weeks	Spain	*Blautia* was negatively correlated with BMI loss percentage in women on a low-fat diet.
Gómez-Pérez et al. 2022 [42]	Mediterranean diet and exercise	(−)	Good	Reduced	16S rRNA, V2–V9 region sequencing	48 weeks	Spain	Decrease in *Blautia* may be associated with the development of non-alcoholic fatty liver disease.
Yuan et al. 2022 [58]	improved ketogenic diet	(+)	Bad	Reduced	16S rRNA, V3–V4 region sequencing	12 weeks	China	Decreased the abundance of *Blautia* and enhanced weight loss in obese individuals.
Wang et al. 2023 [59]	hypocaloric balanced diet	(+)	Bad	Reduced	16S rRNA, V3–V4 region sequencing	12 weeks	China	Diet led to significant weight loss and changes in the gut microbiota of obese individuals, including a decrease in the abundance of *Blautia*.
Medawar et al. 2021 [60]	eating habits and Roux-en-Y gastric bypass	(−)	Good	nd	16S rRNA, V3–V4 region sequencing	NA	Germany	*Blautia* abundance correlated with healthier eating behavior, but this was reduced with Roux-en-Y gastric bypass.
Wang et al. 2023 [56]	improved ketogenic diet and exercise	(−)	Good	nd	metagenomic sequencing	12 weeks	China	*B. obeum* was positively associated with VFA, while a negative association was observed with *B. producta*, *B. hansenii*, *B. wexlerae*, and *Blautia* sp. CAG257.
Pataky et al. 2016 [57]	hypocaloric hyperproteic diet	(−)	Good	nd	shotgun metagenomics	3 weeks	Switzerland	*Blautia* was negatively associated with changes in the body fat mass.

nd: not determined, NA: not applicable, (−): negative association, and (+): positive association.

## Data Availability

The original contributions presented in the study are included in the article/Appendix A, and any further inquiries can be directed to the corresponding author/s.

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
