# Peer review of "The Ambiguous Correlation of Blautia with Obesity: A Systematic Review"

_microorganisms, 2024, doi:10.3390/microorganisms12091768_

Round 1

Reviewer 1 Report

Comments and Suggestions for Authors

According to the manuscript titled "The ambiguous correlation of Blautia with obesity: A systematic review" by Warren Chanda and colleagues. A global epidemic of obesity poses significant health and economic challenges due to its complex and multifactorial nature. In addition to diet and lifestyle, the gut microbiota is increasingly recognized as a contributor to obesity development. A number of studies have reported both potential probiotic properties and casual factors for obesity associated with Blautia, one of the major intestinal bacteria of the Firmicutes phylum. The purpose of this systematic review is to summarize current understanding of the Blautia-obesity relationship and to evaluate evidence from animal and clinical studies, in order to inform future research and therapeutic strategies targeting the gut Blautia in obese patients. In regard to the present manuscript, I would like to make a few comments.

  • The manuscript should introduce terms related to gut microbiota

  • It is possible that the manuscript is a systematic review. According to my opinion, the introduction is too long in conjunction with the table

  • There are several steps that are missing in the present manuscript regarding the material and methods of the systematic review. It is important to understand the search equation, the PICOS criteria, the bias evaluation, and how the articles were selected or assessed. It is possible to accomplish this using several tools, such as JBI guidelines or RevMan

  • There is a division of the results section according to the manuscripts found in the literature. Perhaps this should be explained in the introduction to facilitate understanding.

  • Discussion is excellent; however, the narrative review should be changed to a systematic review in accordance with PRISMA guidelines.

  • From my point of view, the manuscript should be reorganized in the introduction, describing the next sections of the results. Add several key aspects to the material and methods, and emphasize more of the key points in the results and discussion

Author Response

  • Comment 1: The manuscript should introduce terms related to gut microbiota

Response. Thank you. Some terms such as probiotics (Line 242), prebiotics (Line 263) and SCFAs (Line 55), including gut microbiota (Line 48) have been introduced.

  • Comment 2: It is possible that the manuscript is a systematic review. According to my opinion, the introduction is too long in conjunction with the table

Response. Thank you. We have revised the introduction section (Line 30-74). To further shorten the introduction, Table 1 is changed to a supplementary Table S1 (Line 61, 87)

  • Comment 3: There are several steps that are missing in the present manuscript regarding the material and methods of the systematic review. It is important to understand the search equation, the PICOS criteria, the bias evaluation, and how the articles were selected or assessed. It is possible to accomplish this using several tools, such as JBI guidelines or RevMan

Response. Thank you. The Method section has been rearranged in line with the PRISMA guidelines. It consists of literature search strategy (Line 94-99), study selection with the PICOS criteria (Line 106-113), data extraction (Line 115-121), Bias evaluation (Line 124-130), and data synthesis (Line 132-133). Because we wanted to understand the effect of medical treatment or lifestyle interventions on gut microbiota with the outcome variables (Blautia population vs obesity status), we included all studies that met the inclusion criteria for data extraction, irrespective of their quality (part of JBI guidelines). The risk-of-bias domains considered were selection bias (“Are the groups comparable such that an observed difference is likely attributable to the treatment rather than a confounder?”), performance bias (“Was the approach to husbandry the same for all treatment groups and was caregiving done without knowledge of the treatment group?”), and detection bias (“Was the approach to assessing the outcomes the same in both groups and done without knowledge of the group?”) as outlined in an article, “Annette M. O'Connor, Jan M. Sargeant, Critical Appraisal of Studies Using Laboratory Animal Models, ILAR Journal, Volume 55, Issue 3, 2014, Pages 405–417, https://doi.org/10.1093/ilar/ilu038”.

  • Comment 4: There is a division of the results section according to the manuscripts found in the literature. Perhaps this should be explained in the introduction to facilitate understanding.

Response. Thank you. The last paragraph of the introduction has been revised to effectively summarize the objectives of the systematic review, to provide a clear overview of the research focus and the potential implications of the findings. It has highlighted the divisions of the results sections (Line 68-74). Also, a statement has been added to explain the divisions (Line 176-180)

  • Comment 5: Discussion is excellent; however, the narrative review should be changed to a systematic review in accordance with PRISMA guidelines.
  • Thank you. The requirements have been met. For instance, on the PRISMA checklist, 23a about the general interpretation of the results has been covered in Lines 407-476, 23b&c about limitations of the evidence and the review process has been included in Lines 477-486, while 23d about the implications of the results for practice, policy, and future research has been included in Lines 487-507.
  • Comment 6: From my point of view, the manuscript should be reorganized in the introduction, describing the next sections of the results. Add several key aspects to the material and methods, and emphasize more of the key points in the results and discussion

Response.  Thank you. The introduction has been reorganized in a way that the last paragraph (Line 68-74) indicates the main objective (Blautia’s role in obesity) and specific objectives that include 1) exploring the abundance of Blautia in the gut microbiome of obese individuals concerning any treatment and lifestyle interventions. 2) exploring how Blautia populations respond to any treatments and lifestyle changes in obese individuals, and 3) examining associations between changes in Blautia abundance and the efficacy of employed interventions in managing obesity. These specific objectives are the basis for the divisions in the results section. Section 3.2 addresses objectives 1 and 2, while section 3.3 addresses objective 3. Both the results and discussion sections have emphasized on the 3-outlined objectives. The method section has been re-arranged following the PRISMA guidelines (see response for comment 3)

Reviewer 2 Report

Comments and Suggestions for Authors

Obesity is a complex and multifactorial disease with global epidemic proportions, posing significant health and economic challenges. Whilst diet and lifestyle are well-established contributors to the pathogenesis, the gut microbiota's role in obesity development is increasingly recognized. Blautia, as one of the major intestinal bacteria of the Firmicutes phylum, is reported with both potential probiotic properties and casual factors for obesity in different studies making its role controversial.

The writing is generally clear and understandable, but there are a few grammatical errors and awkward phrasings that could be improved. For instance, "casual factors" should be corrected to "causal factors."

Figure 1 does not seem to be explained in the text, nor is the content of figure 1B commented on. 

Author Response

  • Comment 1: The writing is generally clear and understandable, but there are a few grammatical errors and awkward phrasings that could be improved. For instance, "casual factors" should be corrected to "causal factors."

Response. Thank you. It has been revised (Line 17).

  • Comment2: Figure 1 does not seem to be explained in the text, nor is the content of figure 1B commented on.

Response. Thank you. Fig. 1A shows health conditions that can occur or exacerbated by obesity. Line 33-39 refers to Fig. 1A in text. Fig. 1B indicates the detection of the Blautia in various phenotypes. We have included in the text Line 61-62; and it is further cited in Line 422.

Round 2

Reviewer 1 Report

Comments and Suggestions for Authors

Thank you for taking into account my previous comments and rearranging your manuscript into a systematic review. I have no further comments to make.